# Photocatalytic Degradation of Deoxynivalenol over Dendritic-Like α-Fe_2_O_3_ under Visible Light Irradiation

**DOI:** 10.3390/toxins11020105

**Published:** 2019-02-11

**Authors:** Huiting Wang, Jin Mao, Zhaowei Zhang, Qi Zhang, Liangxiao Zhang, Wen Zhang, Peiwu Li

**Affiliations:** 1Oil Crops Research Institute of the Chinese Academy of Agricultural Sciences, Wuhan 430062, China; wanghuitingsdau@163.com (H.W.); maojin106@whu.edu.cn (J.M.); zwzhang@whu.edu.cn (Z.Z.); zhangqi521x@126.com (Q.Z.); zhanglx@caas.cn (L.Z.); zhangwen@oilcrops.cn (W.Z.); 2Key Laboratory of Biology and Genetic Improvement of Oil Crops, Ministry of Agriculture, Wuhan 430062, China; 3Key Laboratory of Detection for Biotoxins, Ministry of Agriculture, Wuhan 430062, China; 4Laboratory of Risk Assessment for Oilseeds Products (Wuhan), Ministry of Agriculture, Wuhan 430062, China; 5Quality Inspection and Test Center for Oilseeds Products, Ministry of Agriculture, Wuhan 430062, China

**Keywords:** deoxynivalenol, degradation, photocatalysis, α-Fe_2_O_3_, degradation products

## Abstract

Deoxynivalenol (DON) is a secondary metabolite produced by *Fusarium*, which is a trichothecene mycotoxin. As the main mycotoxin with high toxicity, wheat, barley, corn and their products are susceptible to contamination of DON. Due to the stability of this mycotoxin, traditional methods for DON reduction often require a strong oxidant, high temperature and high pressure with more energy consumption. Therefore, exploring green, efficient and environmentally friendly ways to degrade or reduce DON is a meaningful and challenging issue. Herein, a dendritic-like α-Fe_2_O_3_ was successfully prepared using a facile hydrothermal synthesis method at 160 °C, which was systematically characterized by X-ray diffraction (XRD), high-resolution transmission electron microscopy (HRTEM), scanning electron microscopy (SEM), and X-ray photoelectron spectroscopy (XPS). It was found that dendritic-like α-Fe_2_O_3_ showed superior activity for the photocatalytic degradation of DON in aqueous solution under visible light irradiation (λ > 420 nm) and 90.3% DON (initial concentration of 4.0 μg/mL) could be reduced in 2 h. Most of all, the main possible intermediate products were proposed through high performance liquid chromatography-mass spectrometry (HPLC-MS) after the photocatalytic treatment. This work not only provides a green and promising way to mitigate mycotoxin contamination but also may present useful information for future studies.

## 1. Introduction

Deoxynivalenol (DON) is a water-soluble trichothecene mycotoxin produced by *Fusarium*, which can contaminate many grains such as wheat, barley, corn and other cereal crops [1]. The data from the Food and Agriculture Organization of the United Nations (FAO, 2001) and the European Union (2002), 57% of wheat samples were contaminated by DON in approximately 22,000 tested samples [2]. The international agency for research on cancer (IARC) had classified DON as the third class of carcinogens (not classifiable) in 2002 [3,4]. The contamination of DON can cause enormous economic losses of agriculture and bring serious threats to the health of human beings and animals. It was found that exposure to excessive DON could cause many adverse reactions, including dizziness, nausea and vomiting [5,6]. Therefore, attention has been paid to reduce or mitigate DON using different strategies.

Traditional strategies including physical, chemical and biological ways have been developed to eliminate DON in agro-food and the aqueous environment. The physical ways included washing, density screening, heating, adsorption [7,8,9]. For instance, Pronyk et al. reported that 52% of DON could be reduced after 6 min thermal degradation at 185 °C [10]. Some chemical reagents, such as ozone (O_3_), sodium carbonate (Na_2_CO_3_), sodium bicarbonate (NaHCO_3_) and sodium bisulfite (NaHSO_3_) have been used to remove DON [11]. Ozone treatment has been widely applied in agro-product processing to degrade mycotoxins. Ozone could effectively reduce 53% of DON in wheat in 4 h [12]. In addition, biological technologies have also been applied in DON detoxification [13,14]. Yang demonstrated that the *Lactobacillus plantarum* JM113 could efficiently reduce DON, which was due to the high antioxidant activity of *Lactobacillus plantarum* JM113. [15]. However, due to the stability of DON in agro-food and the environment, these methods often need a strong oxidant, high temperature and more energy consumption. Therefore, exploring the green, efficient, large-scale and environmentally friendly ways to reduce DON is a meaningful and challenging topic.

In recent years, there has been a growing interest in photocatalytic degradation technology of organic pollutants. Compared to above-mentioned ways, photocatalytic degradation has several advantages: environmental-friendly, low cost, and mild conditions [16]. Bai et al. have successfully proved that graphene/ZnO hybrids showed the ability to degrade DON under UV light irradiation [17]. However, UV light accounts for only a small percentage (4%) of solar energy compared to visible light (43%) [18]. Consequently, in order to utilize solar energy more efficiently, it is necessary to develop a visible light responding catalyst for DON degradation. It was found that microcystin-LR could be removed over magnetically N-doped TiO_2_ nanocomposite under visible light irradiation [19]. Our previous studies also presented that two kinds of visible light responding catalysts show high activities for the degradation of aflatoxin B_1_ under visible light irradiation [20,21]. However, to our knowledge, there are only a few reports about the photocatalytic degradation of DON under visible light irradiation.

As an earth-abundant, visible light responding and nontoxic semiconductor material, hematite α-Fe_2_O_3_ with a suitable band gap of ~2.4 eV has received the most attention. Recently, α-Fe_2_O_3_ nanomaterials with different microstructures, particular shapes and particle sizes have been applied to degrade organic pollutants under visible light irradiation, such as methylene blue dye, eosin Y, rhodamine B dyes, tetracycline [22,23,24]. Herein, a visible light responding catalyst, dendritic-like α-Fe_2_O_3_ was prepared facilely by a simple one-step hydrothermal synthesis method at 160 °C. Under visible light irradiation (>420 nm), the dendritic-like α-Fe_2_O_3_ showed superior performance for degradation of DON in aqueous solution compared with commercial α-Fe_2_O_3_. In addition, the possible degradation products after 120 min photoreaction were analyzed and proposed by HPLC-MS.

## 2. Results and Discussion

### 2.1. Crystal Phase and Morphology Analyses

To analyze the crystal phase and crystallinity of the as-prepared product, the XRD pattern of the as-prepared catalyst was presented. As shown in Figure 1a, the XRD pattern of the as-prepared α-Fe_2_O_3_ was consistent with JCPDS No. 01-089-0596, which was a pure phase of rhombohedral α-Fe_2_O_3_. In addition, the representative crystal planes of the rhombohedral α-Fe_2_O_3_ were labeled in the pattern. These planes were ascribed to (012), (104), (110), (113), (024), (116), (122), (214) and (300), respectively. No other peaks were found, indicating that the as-prepared α-Fe_2_O_3_ was pure. Furthermore, the strong and steep peaks indicated that the as-prepared α-Fe_2_O_3_ was of high crystallinity. The XPS was used to estimate the surface chemical compositions and valence state of the as-prepared α-Fe_2_O_3_. From Figure 1b, it was found that Fe and O existed in the as-prepared α-Fe_2_O_3_. In Figure 1c, there are two peaks at 711.18 eV and 724.78 eV, which are consistent with Fe 2p3/2 and Fe 2p1/2 respectively. The two peaks are characteristic peaks of the Fe^3+^ state in Fe_2_O_3_ [25]. The O 1s spectrum showed one peak (529.85 eV in Figure 1d), corresponding to oxygen atoms of the α-Fe_2_O_3_. From the above XRD and XPS results, it could be concluded that the as-prepared product was a pure α-Fe_2_O_3_. 

The microstructure and morphology of the as-prepared α-Fe_2_O_3_ were investigated by SEM and HRTEM, which are presented in Figure 2. The as-prepared α-Fe_2_O_3_ in the SEM image (Figure 2a,b) was a uniform dendritic-like three-dimensional microstructure. The dendritic-like microstructure had branches that stretched in different directions. As shown in Figure 2c, it was found that the α-Fe_2_O_3_ showed the average length of about 2 μm and the average width of the α-Fe_2_O_3_ was about 500 nm. In Figure 2d, the distances of the lattice in α-Fe_2_O_3_ were 0.27 nm, which was consistent with the (014) plane in rhombohedral α-Fe_2_O_3_ structure. The above SEM and HRTEM indicate that the uniform and regular dendritic-like microstructural α-Fe_2_O_3_ was successfully prepared through the facile hydrothermal method.

### 2.2. Photocatalytic Activity

The photocatalytic activities of the dendritic-like α-Fe_2_O_3_ were estimated by the degradation rate of DON in aqueous solution. From Figure 3a, it could be found that the intensity of DON decreased significantly with time over dendritic-like α-Fe_2_O_3_ under visible light irradiation. This indicated that the dendritic-like α-Fe_2_O_3_ had superior photocatalytic activity under the visible light irradiation. In Figure 3b, the blank line revealed that the DON could not be degraded under visible light obviously without a photocatalyst or in the presence of photocatalyst without visible light, which indicated that the catalysts and visible light were necessary for the photocatalytic degradation of DON. In addition, after the suspension was stirred with a magnetic stirring apparatus for 1 h in the dark, the content of a DON standard was reduced slightly. It was found that the dendritic-like α-Fe_2_O_3_ could adsorb more DON than that of commercial α-Fe_2_O_3_, which may enhance the photocatalytic degradation activity. As expected, the dendritic-like α-Fe_2_O_3_ (the blue line in Figure 3b) showed greatly a higher photocatalytic efficiency with a degradation rate of 90.3% in 2 h than that of commercial α-Fe_2_O_3_ (46.7%). There are two reasons why the dendritic-like α-Fe_2_O_3_ had better photocatalytic performance. Firstly, as the three-dimensional nanomaterial, the dendritic-like α-Fe_2_O_3_ is difficult to form aggregation compared with commercial α-Fe_2_O_3_. In addition, the unique morphology of bionic dendritic a-Fe_2_O_3_ tends to increase the absorption of the sunlight and provides more charge transfer pathways and more active sites to improve the efficiency of photocatalytic activities [25].

### 2.3. Degradation Products Analysis

Generally, the photocatalytic degradation products of organic compounds are determined by the kind and quantity of active radicals production during the photoreaction, which may be different in the presence of different catalysts or with different irradiation time [26]. During the photoreaction over α-Fe_2_O_3_, the electrons could be excited and transferred from the valence band (VB) to the conduction band (CB), leaving holes in the VB, which can oxidize the hydroxide ions into hydroxyl radicals (•OH). In addition, in the CB of the α-Fe_2_O_3_, the transferred electrons reacted with dissolved oxygen to form superoxide radical anions (•O_2_^−^). Then, a number of active radicals such as •O_2_^−^ and •OH could react with the active site of DON and form the intermediate products [27,28]. To identify the possible intermediate products of DON, the intermediates after photocatalytic treatment were investigated by HPLC-MS. As shown in Figure 4 the total ion chromatograms (TIC) also revealed that the DON obviously reduced after 2 h photodegradation. A comparison between the initial TIC with that of after photodegradation revealed that two obvious peaks, P1 (*m*/*z* 209.17) and P2 (*m*/*z* 227.0), appeared after the photocatalysis treatment, which may be the intermediate products of DON. The *m*/*z* value of the DON in the positive ESI mode was 297.00. As shown in Figure 5, the possible structures of the two intermediate products were proposed according to previous studies [29,30,31,32]. The products P1 and P2 may have been generated from a series of reactions such as the dehydration, breakage of the carbon chain and reduction of the 12, 13-epoxy group to diene. However, the detailed reaction process needs a deep investigation. As shown in Figure 6, DON is a type B trichothecenes compound and the 12, 13-epoxy group of its structure is associated with its toxicity. In addition, the three hydroxyl groups (-OH) in the DON molecule also are related to its toxicity [33]. It was found that these toxicity sites were destroyed from the proposed structure of the two products, indicating that the toxicity of DON may decrease after efficient photocatalytic treatment over dendritic-like α-Fe_2_O_3_. However, the toxicity of DON products needs more evidence, systematic and long-term evaluations in our future study, which is in progress. From the above results and discussions, it could be deduced that the dendritic-like α-Fe_2_O_3_ have a potential application in the photocatalytic degradation of DON in aqueous solution.

## 3. Conclusions

In conclusion, a light-responsive dendritic-like α-Fe_2_O_3_ with a length of 2 μm and the width of 500 nm was successfully synthesized through a facile hydrothermal synthesis method. The dendritic-like α-Fe_2_O_3_ showed better activity for the degradation of DON in aqueous solution under visible light irradiation compared with the commercial α-Fe_2_O_3_, which was ascribed to the fact that the dendritic-like α-Fe_2_O_3_ was aggregated with difficulty and could provide more charge transfer pathways and more active sites. Most of all, two possible intermediate products, P1 (*m*/*z* 209.17) and P2 (*m*/*z* 227.00), were proposed through HPLC-MS after 2 h photocatalytic treatment. It was found that the main toxicity sites of the 12, 13-epoxy group and hydroxyl groups in DON were destroyed, indicating that the toxicity of DON may decrease after efficient photocatalytic treatment. This work not only provided a new, efficient, green and promising way to reduce mycotoxins such as DON contamination but also presents useful information for future study.

## 4. Materials and Methods 

### 4.1. Materials

The deionized water was from the RIOS16/Milli-Qa10 water purification system (France). Potassium hexacyanoferrate (III) (K_3_[Fe(CN)_6_], >99.5% purity) was purchased from Tianjin Bodi Chemical Reagent Co. (China). For high performance liquid chromatography and high performance liquid chromatography-mass spectrometry analysis, acetonitrile and methanol (mass spectrum grade) were purchased from Fisher Chemical. Formic acid (CH_3_COOH, >95% purity) and DON standard came from Sigma-Aldrich Co. (USA). All chemicals and reagents were used without further purification.

### 4.2. Synthesis of Dendritic-Like α-Fe_2_O_3_ Photocatalyst

In a typical hydrothermal reaction, K_3_[Fe(CN)_6_] (0.528 g) was ultrasonically dispersed (40 kHz, 25 °C, 5 min) in the deionized water (80 mL) and then transferred to a 100 mL Teflon-lined stainless steel autoclave. Then Teflon-lined stainless steel autoclave was put into an oven and heated for 24 h at 160 °C. After it was cooled down to room temperature, the obtained solid was washed with deionized water and methanol for 3 times. The as-prepared catalyst was dried at 60 °C for 6 h. Finally, the dried solid was ground into powder in an agate mortar for later characterization and photocatalytic measurement.

### 4.3. Material Characterization

The morphology and structure of the sample were characterized by X-ray diffraction (Bruker AXS, D8, Germany) with a scanning range from 10 to 70 degree. Scanning electron microscopy (Hitachi-4800, Japan) and transmission electron microscopy (FEI Tecnai G2 F30, America) were used to characterize the particle size and the morphology of the as-prepared sample. X-ray photoelectron spectroscopy analysis was performed on a Kratos XSAM 800 X-ray photoelectron spectrometer with an Mg Kα X-ray source.

### 4.4. Photocatalytic Measurement

The photocatalytic activity test was performed as follows: 0.01 g α-Fe_2_O_3_ was dispersed in 96 mL deionized water through 20 min ultrasonic treatment (40 kHz, 25 °C). Then, 4 mL of DON aqueous solution (100 μg/mL) was added to above α-Fe_2_O_3_ suspension. Then, an α-Fe_2_O_3_ suspension containing DON was magnetically stirred for 1 h in the dark. Then, the suspension was irradiated under visible light (λ > 420 nm) with magnetically stirring at room temperature. The light source was provided by a 300 W Xenon lamp (PLS-SXE 300, Beijing Trusttech Co., Beijing, China). The distance was about 20 cm from the light source to the DON suspension. The suspension after different irradiation time (30 min, 60 min, 90 min, 120 min) were collected to measure the concentration of DON using HPLC (Agilent 1200, USA) equipped with a chromatographic column (Agilent polar C18-A, 5 μm, 150 mm × 4.6 mm). The samples were filtrated through a 0.22 μm filter before detection. The mobile phases were the aqueous solution and acetonitrile in 83:17 (*v*/*v*). The flow rate of mobile phases was 1 mL/min, and the temperature of the column was 35 °C. The excitation and emission wavelengths were 220 nm and 360 nm, respectively. The injection volume was 20 μL. The control and commercial α-Fe_2_O_3_ were performed as the above steps.

### 4.5. Degraded Products Analysis

The high performance liquid chromatography-electrospray ionization mass spectrometry (HPLC-ESI/MS, Thermo Finnigan LTQ XL, USA) was used to analyze the degradation products of DON. The HPLC (Thermo, USA) equipped with a chromatographic column (Syncronis C18, 3 μm, 100 × 2.1 mm, Thermo). The flow rate was 200 μL/min, and the injection volume was 5 μL. The mobile phase included two components. Component A was distilled water with 0.1% formic acid and component B was pure acetonitrile. The ESI was used in positive ion mode and other parameters were as follows: capillary temperature was at 350 °C, and the flow rate of sheath gas, aux gas and sweep gas was at 28 arb, 2 arb, 0 arb, respectively. The pressure of nebulizer was at 4.50 Kv with the capillary voltage at 9 V, tube lens voltage at 95 V; multipole RF Vpp at 400 V. All other parameters in the MS were tuned for maximum signal intensity of a reference solution. The scan ranges of MS were from *m*/*z* = 50 to *m*/*z* = 320.

## Figures and Tables

**Figure 1 toxins-11-00105-f001:**
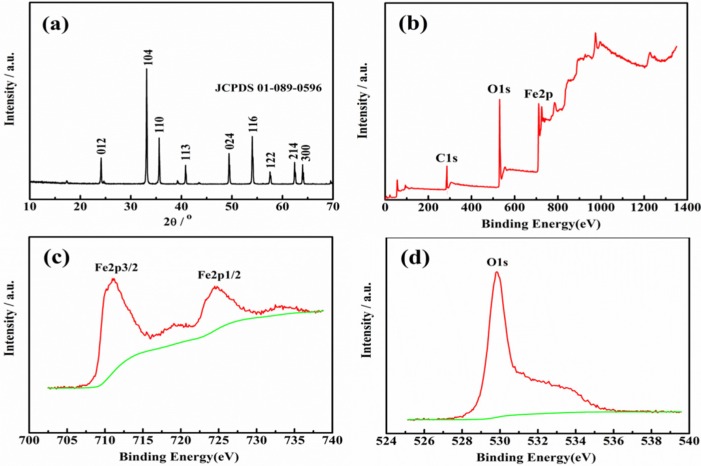
(**a**) XRD pattern of the as-prepared α-Fe_2_O_3_; (**b**) XPS spectra of the as-prepared α-Fe_2_O_3_; High-resolution XPS spectra of Fe (**c**) and O element (**d**).

**Figure 2 toxins-11-00105-f002:**
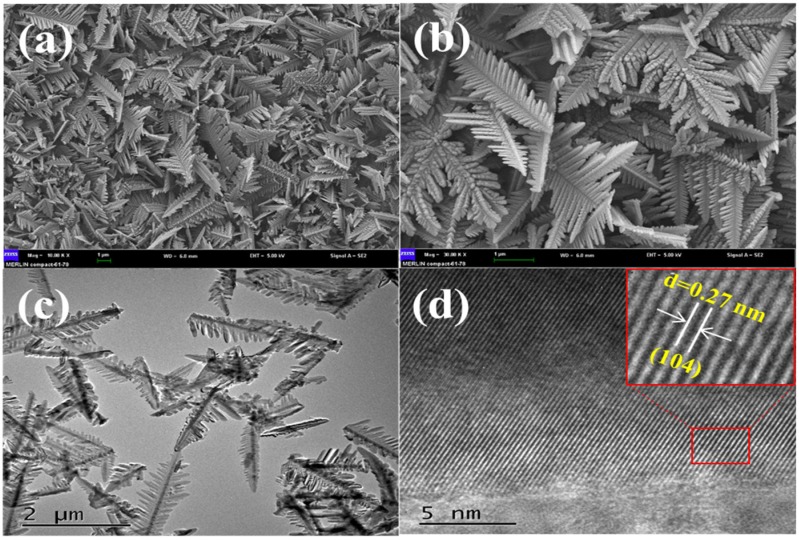
SEM images (**a**,**b**), HRTEM images (**c**,**d**) of the prepared α-Fe_2_O_3_.

**Figure 3 toxins-11-00105-f003:**
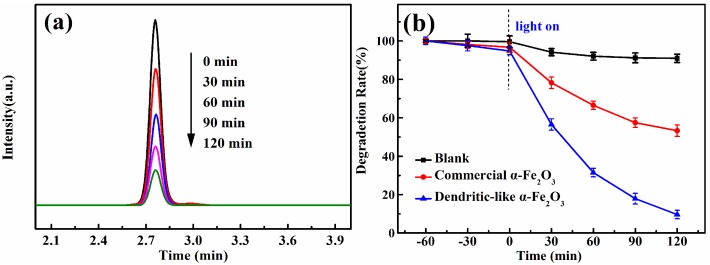
(**a**) HPLC chromatogram of DON photodegradation over dendritic-like α-Fe_2_O_3_ with different times (**b**) Photocatalytic degradation of DON over dendritic-like α-Fe_2_O_3_, commercial α-Fe_2_O_3_ under visible light and blank control.

**Figure 4 toxins-11-00105-f004:**
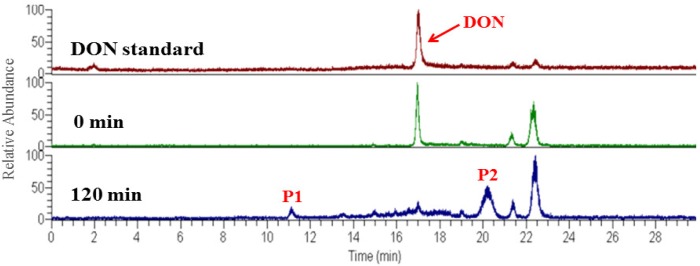
TIC of the sample for the DON standard, before reaction (0 min) and after 120min reaction.

**Figure 5 toxins-11-00105-f005:**
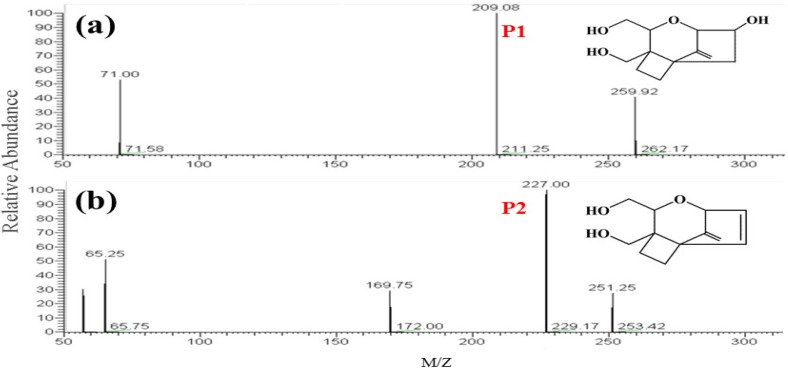
The full scan mass spectrums and possible structures of P1 (**a**), P2 (**b**).

**Figure 6 toxins-11-00105-f006:**
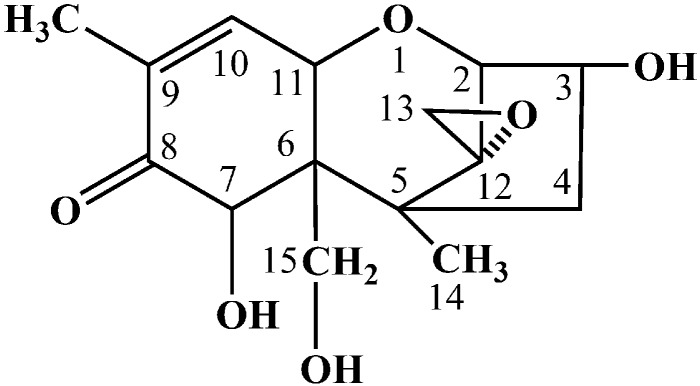
Chemical structure of DON.

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
