# Peer review of "Photocatalytic Degradation of Deoxynivalenol over Dendritic-Like α-Fe2O3 under Visible Light Irradiation"

_toxins, 2019, doi:10.3390/toxins11020105_

Round 1
Reviewer 1 Report
In the present study, the authors have developed a degradation model using dendritic-like α-Fe2O3 and has been applied to degrade DON, a fungal toxin contaminates several edible crops. Although the experimental design seems to be sound and the results are interesting, the degradation products still need to be identified. Also, the toxicity of these products should be tested to prove that the model is fit for purpose. From the experience, degradation of food contaminates can produce more toxic products which render the degradation process useless. Therefore, I suggest that the authors perform the proposed experiment below to be sure of the identity of the products. Further, to test the toxicity of the products and combined to the current work or consider of an additional separated publication.
Abstract
The abstract needs to be rewritten again. It contains mistakes (both Scientifically and grammatically). DON is not carcinogenic.
This sentences “As a main water-soluble mycotoxin with high toxicity and carcinogenicity, wheat, barley, corn and their products are susceptible to contamination of DON” is not correct at all. Wheat, corn and barely are susceptible to by contaminated by DON because the producing fungus, fusarium species, is attacking these crops. Being a water-soluble chemical substance has NO connection with the pathological events exerted by the fungus.
What are the products of the degradation process ? have you tested the toxicity of these products in order to be sure that the degradation products are safe or at least less than the parent compounds ?
Introduction
Ø Line 28, please remove the word ‘so on’ and replace it with the “other cereal crops”
Ø Line 29, please replace the reference number 2 with the original reference stating the mentioned figure in your manuscript “57% of wheat samples were contaminated by DON in approximately 22000 tested samples”
Ø Line 31, you have stated that DON is carcinogenic, however in fact the toxin is not carcinogenic is “Group 3 (not classifiable) human carcinogen due to inadequate evidence of animal carcinogenicity”. Please remove the reference number 3 as it is not sutiable at all and replace it with the following “De Ruyck et al., 2015 Dietary mycotoxins, co-exposure, and carcinogenesis in humans: Short review” and Abdallah et al., 2015 Occurrence, Prevention and Limitation of Mycotoxins in Feeds”.
Ø Line 46, reference number 13 is for AFB1 detoxification not for DON. Please replace it with “Vanhoutte et al., 2017 Microbial detoxification of deoxynivalenol (DON), assessed via a Lemna minor L. bioassay, through biotransformation to 3-epi-DON and 3-epi-DOM-1”
Material and method & results and discussion
The author showed the efficacy of the dendritic-like α-Fe2O3 to degrade DON using LC-MS/MS, however they did not show the identity of the products (using only full scan is not enough). Further they relayed on the previous studies to show the identity of the products. In conclusion, using only the full scan mode and mention the mass of the compound indeed is not enough. I would suggest to use HRLC-MS/MS and try to fragment this product to be sure of the identity. After you identify these two products, if there are commercial standards available for them then it would be the best to inject the standard and the treated samples into LCMS/MS and then you are 100% sure of the identity (in case there are commercial standards).
Indeed, testing the toxicity of these products is a must in order to confirm the efficacy of the developed dendritic-like α-Fe2O3.
Ø Figure number 6 , it is “chemical structure” NOT “structures”, please remove the “s”
Author Response
Dear reviewers,
Thank you for your nice and careful review, please see our response to your comments in the attachment. The editors give us 10 days to update our manuscript, and we hope that the revised manuscript can be aligned with your requirements.
Best regards and happy Chinese New Year!

Reviewer 2 Report
The work presented by Wang and co-workers is within the scope of the journal, and sound. It describes the photocatalytic degradation of DON in aqueous solutions. It was tested in a model system (aqueous solution), and the efficacy seems to be related to the presence of water (production of hydroxyl radicals). This is obviously a preliminary and promising work, but it is not clear how this can be exploited in a real food system.
As a general comment, I ask authors for a deep English revision of their work. I am also not a native speaking English, but I do believe to text is not clear, and there are many typing and grammar issue to correct. I will present a few examples below.
- First sentence of Abstract (and in many other places): authors refer to DON as a kind of trichotechene. We may simply refer to DON as a trichothecene (remove ‘kind of’). Also in abstract, it is said that a kind of dendritic-like α-Fe2O3 was prepared. Also in this case, the use of the ‘kind of’ can be omitted. In the whole paper, “… kind of …” is improperly used many times.
- In abstract and in Results, the degradation of DON is reported as 90.37%. this is a very precise value (too many significant algharisms). Why not just 90%?
- Introduction, line 31-32: It is said that DON has high toxicity and carcinogenicity, being classified by IARC in group 3. It is true that DON is classified in group 3; meaning that it is not classifiable as to its carcinogenicity to humans. So, there is no evidence on its carcinogenicity to humans.
- Page 2: Lactobacillus plantarum; plantarum is not with capital letter.
- page 4, line 115: not absorb, but adsorb
- page 5-6, lines 168-171: this sentence is not clear.
Author Response
Dear reviewers,
Thank you for your nice and careful review, please see our response to your comments in the attachment. We hope that the revised manuscript can be aligned with your requirements.
Best regards and happy Chinese New Year!

Round 2
Reviewer 1 Report
the reviewers have responded to all the comments. No further comments are raised by me.